# Alternatives to Carbon Dioxide—Taking Responsibility for Humanely Ending the Life of Animals

**DOI:** 10.3390/ani9080482

**Published:** 2019-07-24

**Authors:** Shannon Axiak Flammer, Chantra Eskes, Ingrid Kohler, Awilo Ochieng Pernet, Peter Jakob, Michael Marahrens, Thomas C. Gent, Huw Golledge, Dan Weary

**Affiliations:** 1Department of Clinical Veterinary Medicine, Section of Anesthesia and Analgesia, Vetsuisse Faculty, University of Berne, Laenggassstrasse 124, 3012 Bern, Switzerland; 2Directorate Swiss 3R Competence Centre, Hochschulstrasse 6, 3012 Bern, Switzerland; 3Division of Laboratory Animal Welfare, Federal Food Safety and Veterinary Office, Schwarzenburgstrasse 155, 3003 Bern, Switzerland; 4Division of International Affairs, Federal Food Safety and Veterinary Office, Schwarzenburgstrasse 155, 3003 Bern, Switzerland; 5Division of Food Hygiene and Nutrition, Federal Food Safety and Veterinary Office, Schwarzenburgstrasse 155, 3003 Bern, Switzerland; 6Institute of Animal Welfare and Animal Husbandry, Friedrich-Loeffler-Institut, Dörnbergstraße 25/27, 29223 Celle, Germany; 7Department of Clinical and Diagnostic Services, Section of Anaesthesiology, Vetsuisse Faculty, University of Zurich, Winterthurerstrasse 258c, 8057 Zurich, Switzerland; 8Universities Federation for Animal Welfare (UFAW), The Old School, Brewhouse Hill, Wheathampstead, Hertfordshire AL4 8AN, UK; 9Animal Welfare Program, University of British Colombia, 2357 Main Mall, Vancouver, BC V6T 1Z4, Canada

**Keywords:** carbon dioxide, euthanasia, laboratory animals

## Abstract

**Simple Summary:**

Carbon dioxide has long been considered one of the better methods for euthanizing laboratory rodents because it allows termination of several animals at one time, does not require handling of the animal, is easy to use, is inexpensive, and is environmentally friendly. Research, though, has shown that this gas is aversive to rodents and that it may be inhumane to expose them to this gas while they are conscious. Therefore, the Swiss Federal Food Safety and Veterinary Office has set out to find a suitable replacement and organized a meeting that included representatives and experts of the different stakeholders involved in this process in order to find a solution. The conclusion of this meeting was that a replacement is required, and the next step would be to draft a research strategy to find a suitable replacement.

**Abstract:**

Carbon dioxide (CO_2_) is commonly used to kill rodents. However, a large body of research has now established that CO_2_ is aversive to them. A multidisciplinary symposium organized by the Swiss Federal Food Safety and Veterinary Office discussed the drawbacks and alternatives to CO_2_ in euthanasia protocols for laboratory animals. Dialogue was facilitated by brainstorming sessions in small groups and a “World Café”. A conclusion from this process was that alternatives to CO_2_ were urgently required, including a program of research and extension to meet the needs for humane killing of these animals. The next step will involve gathering a group of international experts to formulate, draft, and publish a research strategy on alternatives to CO_2_.

## 1. Introduction

Carbon dioxide (CO_2_), rendering animals unconscious and causing death from brain acidosis, is widely used for euthanasia of animals used in research [1] and for the stunning of livestock animals [2]. This procedure is reliable and inexpensive, but a growing body of research has shown that CO_2_ is highly aversive to rodents, causing anxiety, fear, and pain, raising doubts about its use as a method for humane euthanasia [3].

High concentrations (>40%) of CO_2_ cause pain in humans [4], and likely also in rodents based upon similarities in nociceptors in nasal and ocular epithelia. Research has shown that gradually increasing CO_2_ concentrations (e.g., 20% of chamber volume per minute), as is mandatory in the European Union [5], can reduce the likelihood that these animals will experience pain before they are fully anesthetized. However, even low concentrations of CO_2_ cause anxiety in both humans [6] and mice [7,8], and rats will expose themselves to a ‘threatening’ environment in order to avoid exposure to even low concentrations of CO_2_ [9,10]. Panic/defense-related brain circuits, such as the hypothalamic-pituitary axis, are activated in rats after brief exposure to 20% CO_2_ [11]. Furthermore, mice may be especially capable of detecting atmospheres with even low concentrations of CO_2_ because of their well-developed olfactory system that can detect subtle changes in atmospheric CO_2_ and oxygen concentrations [12,13]. Therefore, CO_2_ exposure cannot be considered a humane method for killing rodents because it is likely to cause pain at high concentrations and fear or panic in low concentrations.

In response to this growing body of evidence showing that CO_2_ exposure is harmful for animal welfare, the Swiss Federal Food Safety and Veterinary Office (FSVO) hosted a symposium designed to discuss the mechanisms of action, associated drawbacks, and alternatives to inhaled CO_2_ as a method of killing laboratory animals. The objective of this symposium was to critically discuss the criteria for a good death for animals and consider what killing methods meet these criteria.

## 2. Listening and Discussing—Gathering the Perspectives of Different Stakeholders

To ensure wide-ranging views, email invitations were sent out to stakeholders with a broad range of professions and expertise. A total of 117 people attended; the list of represented institutions and fields can be found in Box 1.
Box 1Participants in the symposium, “Alternatives to Carbon Dioxide”, came from the following institutions, organizations, and professions.Researchers from academia, higher education institutions, and the pharmaceutical industrySwiss animal welfare officersSwiss 3R Competence CentreGerman Centre for the Protection of Laboratory Animals (Deutsches Zentrum zum Schutz von Versuchstieren)Pig and poultry slaughtering, meat industryVeterinary anesthesia expertsLaboratory animal medicine expertsSwiss laboratory animal science teaching institutionsSwiss cantonal veterinary officesSwiss cantonal animal experimentation committeesSwiss Federal Commission for Animal ExperimentationSwiss Federal Office of Public HealthGerman, Norwegian, and Danish governmental animal welfare institutions for animal experimentation and slaughteringSwiss and international animal welfare organizations

Participants were reminded that Swiss animal welfare law requires that animals be slain with care, consideration, and without suffering [14]. The law also requires that only essential procedures are carried out on animals and that those procedures be continuously reexamined.

### 2.1. What is a Good Death for Animals?

Five questions were designed to stimulate the participants to consider what constitutes a good death for animals (Box 2). During a moderated plenary session, the participants were allowed 5 min to discuss each question with their neighbors. The floor was opened to provide responses that subsequently were summarized by the moderators. Eight criteria for a good death for animals were defined in this plenary discussion (Box 3).
Box 2Five questions designed to stimulate group discussion on what constitutes a good death for animals.What are your first thoughts in reaction to the question ‘What is a good death for animals?’How do you know that an animal is suffering?What makes up a ‘good death’ from the perspective of the animal (that is being killed) and the human (that kills the animal)?Is a ‘good death’ different depending on the purpose or environment? [For example, livestock (slaughter versus companion animals versus experimental animals (termination of the study /destruction of surplus animals)]?Which criteria would you include in a checklist for ‘What is a good death for animals’?
Box 3Criteria describing “a good death for animals” based on the responses of 117 experts of various disciplines following a dynamic interactive session on animal death.A good death for animals:is not associated with suffering, pain, anxiety, stress, or distressis reliable and under control and expert supervision until deathis immediate, without delay, and irreversibleis safe for the staff and other animalsavoids contamination of the environment

### 2.2. Alternatives to Carbon Dioxide

Following this discussion, three experts on laboratory animal euthanasia (Huw Golledge, Dan Weary, and Thom Gent) presented on the use of CO_2_ and other gases for euthanasia of rodents. It is possible that other inhaled euthanasia agents might be considered a refinement over CO_2_ for euthanasia of laboratory rodents. Evidence exists from several experimental approaches designed to allow stronger inferences regarding aversion to different agents. One approach is the use of “approach-avoidance” tests. In this method, mice and rats are trained to enter a chamber for a highly desired food reward (in some of the rat work, for example, researchers use 20 “Honey Nut Cheerios”, a sweet breakfast cereal). In baseline conditions, animals always ate all 20 Cheerios, taking about 5 min to do so. Once they were fully trained, the chamber was gradually filled with CO_2_ while the rodent was still eating, and the researchers measured the time spent in the chamber and the number of Cheerios consumed. In this experimental design, an ideal agent can be considered one where the animal would not escape, but rather stay in the chamber eating Cheerios until fully anesthetized. When exposed to CO_2_, all rats escaped the chamber after consuming just a few Cheerios, leaving when concentrations reach just ~12%, far below the level required to render the animal unconscious [15]. Across a series of studies involving 52 subjects, no rat or mouse ever remained in CO_2_ long enough to lose consciousness. In contrast, when inhalant anesthetics (sevoflurane or isoflurane) were introduced into the chamber, approximately one-third of naïve animals tested stayed in the chamber until they lost consciousness [15,16,17,18], but rats that had previous experience with inhalant anesthetics showed aversion responses more similar to those exposed to CO_2_ [10].

A second experimental approach reviewed was the use of “aversion-avoidance” testing. This method is based upon the idea that rodents prefer a dark environment. The test uses a specially designed chamber that provides the animal a choice between light and dark compartments. Animals are introduced into the chamber and are free to move from side to side, but typically settle down and spend the majority of their time in the dark compartment. In this experimental design, an ideal agent can be considered one that the animal would not escape, and thus stays in the dark chamber exposed to the agent until fully anesthetized, rather than escaping to the aversive bright chamber. When exposed to inhalant anesthetics, approximately 2/3 of naïve mice and rats remained there until they become unconscious, rather than moving into the anesthetic-free but brightly lit chamber. In contrast, when exposed to CO_2_, all animals fled and exposed themselves to the aversive light chamber [9,10,19]. Based on this evidence, inhalant anesthetics can be considered a refinement over CO_2_ for rendering naïve mice and rats unconscious. Once fully anesthetized, animals can be killed using an overdose of the gas anesthetic or via some secondary method.

Inert gases may also have a role in the refinement of gas euthanasia of mice. The effects of nitrogen (N_2_), helium, argon, and xenon (Xe) have been compared to those of CO_2_. Differences were observed in the number of animals displaying epileptiform activity in their electroencephalogram when exposed to these gases, the seizure time after loss of motion, and the seizure time before brain death [20]. The results suggest that epileptiform activity during exposure to inert gases is likely to result from hypoxia, but occur close to brain death, following loss of consciousness, and are therefore unlikely to impact upon the welfare of animals. Furthermore, mice undergoing single-gas euthanasia with Xe did not show jumping or freezing behaviors and had reduced locomotion compared to baseline, in contrast to CO_2_ which resulted in behavioral excitation. Electroencephalographic recordings revealed sedative effects from Xe and N_2_ but heightened arousal from CO_2_ [21,22]. It must be noted that these results are based on mice and could be different for rats.

### 2.3. A World Café

After these lectures, a brainstorming session was undertaken in a ‘World Café’ format. Tables with large pieces of paper and markers were organized. A moderator was present at each table and one of three questions were posed to the participants (Box 4). The moderator encouraged discussion and participants were asked to write down their thoughts and ideas. All participants rotated through all three questions in groups of seven to eight.
Box 4World Café questions used to stimulate brainstorming on alternatives to carbon dioxide for euthanasia and slaughter.What would you do if you could not use carbon dioxide today for euthanasia?Can you suggest any novel methods or novel refinements to existing methods (not covered by the talks today) that might be a humane and practical alternative to carbon dioxide?How can your professional experience contribute to the search of alternatives to carbon dioxide?

## 3. Identified Roadblocks

In response to the question “What would you do if you could not use CO_2_ today for euthanasia?” participants stated they would use physical methods (such as decapitation), inhalant anesthetics, or a combination of different methods (for example, anesthetics to render the animal unconscious, followed by a physical method for killing). The choice of the method would depend on considerations such as purpose, number of animals, species, and age.

In addition to these suggestions, some participants made other suggestions for possible refinements including an opioid overdose, microwave heating of the brain, single-pulse ultra-high current stunning, low atmospheric pressure stunning (LAPS), freezing of very small animals in liquid nitrogen, captive bolt, maceration of small animals, and cervical dislocation.

Some participants also discussed whether CO_2_ delivery methods could be refined, although the consensus position (supported by current research) was that no method of delivery would render this method humane. However, better handling and delivery methods could be helpful for use with alternative methods, including anesthetic gases. For example, distress may be reduced by conducting procedures in the home cage, with known cage mates, to reduce handling distress before the procedure.

Several interesting ideas were suggested when participants reflected on their own professional experience and how this could contribute to the search for alternative methods to CO_2_. These suggestions included:Better use of technology to identify and define parameters that occur during death (e.g., behavior, physiological changes) that are species and category dependent.Finding ways of reducing the number of animals that need to be killed. For example, many laboratory animals are killed simply because these are deemed as surplus animals from the breeding program. Therefore, continued refinement of the breeding program, combined with better programs to match animals and researchers (e.g., AniMatch (www.swiss.animatch.eu)), need to be encouraged.Undertaking a comprehensive evaluation of the entire infrastructure from housing type to transportation to limit stress from the beginning until the end of an animal’s life.Reflection on the practice of research as an entity, from experimental design and grant application to publication of results. For example, there is pressure on scientists to obtain grants and publish the results as soon as possible. Several steps in this process (ethical evaluation, review, and decision to award a grant or publish a manuscript) depend on a severely taxed voluntary peer review system. For many scientists performing animal experiments, there is little time, funding, or incentive to adopt refinements in their procedures, including procedures used for killing.Finally, participants called for more research on humane killing methods and societal acceptance of these procedures (as part of more general move towards more transparency regarding animal use).

Perhaps the most valuable contribution from the event was the combination of ideas, sense of community, and positive energy generated from this diverse, multidisciplinary group of people. Each individual brought a range of perspectives, respect for the research conducted to date, and an openness for change. The World Café approach was highly valued, and we suggest this as an effective model for addressing contentious issues in science policy.

## 4. Next Steps

Based on the results described here, the FSVO is proposing to find alternatives to CO_2_ for killing of laboratory rodents (mice and rats). A group of international experts will be gathered to draft a research strategy on alternatives to CO_2_. This research strategy draft will be presented at the 2019 3R Symposium on alternatives to CO_2_ and all participants will be invited to provide feedback on the draft. The finalized research strategy will be published.

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
