# Peer review of "Alternatives to Carbon Dioxide—Taking Responsibility for Humanely Ending the Life of Animals"

_animals, 2019, doi:10.3390/ani9080482_

Round 1

Reviewer 1 Report

I highly recommend this paper be accepted as-is. The relevance and importance of the key issues outlined in this symposium are very high with special regard to the animal and human pain and distress caused by CO2 methods of euthanasia. I commend the efforts put forth by the authors in rallying an excellent stakeholder representation in the discussion. I look forward to the next step outcomes in finding superior alternatives to CO2 euthanaisa of animals in research.

Author Response

Dear Reviewer:

Thank you very much for taking the time to review our manuscript and for the positive feedback.

Best regards,

The author

Reviewer 2 Report

Simple Summary:

Line 28 (&49) - carbon dioxide is not safe for humans, workplace exposure limits are low (15000ppm short term), substantially below the use of CO2 in stunning/killing systems. There have been a number of recorded deaths due to accidents in and around CO2 slaughter systems.

Line 29 - "recent" research - whilst there has been increased highly detailed work reported in the last few years, the aversive nature of carbon dioxide has been researched for 25 years. 

line 92 - "sacrificed" implies some religious aspect - only a translation issue.

Line 93 "indispensable burdens" not clear , perhaps "...only essential procedures are carried out on animals, and that those procedures be continuously re-examined"

Reviewer 3 Report

This is a very good preliminary process to find out a refinement for CO2, and a timely needed topic to discuss.

Introduction- Authors explained distress and aversion to CO2 only in rodents (mice and rats). But the topic is included animals. If the objective is finding a humane killing method for all animals, then authors should provide evidence of aversion and distress associated with CO2 in other species in the introduction. There are many more research have been conducted in poultry species and other livestock species. line 57-63 explained all about mice and rats and no information related to other species.

The objective need to be clearly identified in the manuscript. e.g. whether all animals or rodents only.

Line 141- Better to replace animals with naïve rats and mice. Reader is confusing what animals have been used in this study. Authors have explained animals were rat and mice in the Line 146.Better to introduce rat and mice at the beginning of this paragraph.

 Authors also explained the possibility of using inert gases only in mice and no evidence in other species.

Conclusion- In conclusion authors try to generalize results to all animals. But the explained results do not clearly indicate that CO2 is distress to all species.

In overall, the methodology and the results need to be explained in detail focusing to the objective of this study.CO2 is highly differed in aversion and distress in different species.Different CO2 concentrations and different filling rates also results varied outcomes in different species. Thus it is not good to generalize findings of mice and rats to all other species.
